# Metastasis, an Example of Evolvability

**DOI:** 10.3390/cancers13153653

**Published:** 2021-07-21

**Authors:** Annick Laruelle, Claudia Manini, Elena Iñarra, José I. López

**Affiliations:** 1Department of Economic Analysis, University of the Basque Country (UPV/EHU), 48015 Bilbao, Spain; annick.laruelle@ehu.eus (A.L.); elena.inarra@ehu.eus (E.I.); 2IKERBASQUE, Basque Foundation of Science, 48011 Bilbao, Spain; 3Department of Pathology, San Giovanni Bosco Hospital, 10154 Turin, Italy; claudia.manini@aslcittaditorino.it; 4Public Economic Institute, University of the Basque Country (UPV/EHU), 48015 Bilbao, Spain; 5Department of Pathology, Cruces University Hospital, 48903 Barakaldo, Spain; 6Biocruces-Bizkaia Health Research Institute, 48903 Barakaldo, Spain

**Keywords:** cancer, metastasis, genomic analysis, microenvironment, tumor ecology, game theory

## Abstract

**Simple Summary:**

Cancer is a complex disease. Modern molecular technologies are progressively unveiling its genetic and epigenetic complexity, but still many key issues remain unknown. Considering cancer as a social dysfunction in a community of individuals has provided new perspectives of analysis with promising results. This narrative considers both approaches with respect to the metastatic process, the final cause of death in most patients affected by this disease.

**Abstract:**

This overview focuses on two different perspectives to analyze the metastatic process taking clear cell renal cell carcinoma as a model, molecular and ecological. On the one hand, genomic analyses have demonstrated up to seven different constrained routes of tumor evolution and two different metastatic patterns. On the other hand, game theory applied to cell encounters within a tumor provides a sociological perspective of the possible behaviors of individuals (cells) in a collectivity. This combined approach provides a more comprehensive understanding of the complex rules governing a neoplasm.

## 1. Introduction

Modern treatment modalities are obtaining longer survivals and even cure in a significant percentage of patients with cancer, transforming the disease into a chronic-like condition. However, cancer still remains today the leading cause of death in Western countries. Metastatic dissemination is responsible for more than 90% of tumor deaths and is a challenge for modern oncology. However, metastasis is an extremely inefficient process. It has been estimated that only 0.01% of circulating tumor cells succeed in developing it [1]. The acquisition of motility by tumor cells involves a complex self-organized and self-regulated systemic cellular organization emerging from Hopfield-like dynamics [2]. These dynamics tightly regulate cell migration and other critical functions like associative memory in the cell, as it has been demonstrated very recently in unicellular organisms like amoebae [3].

The initialization of the metastatic process is possible only under regulated cellular metabolic conditions. These particular conditions allow the occurrence of epithelial-to-mesenchymal transition processes that enable epithelial cells to acquire amoeboid motility through the development of a cascade of molecular events. Unveiling the intricate interactions between tumor cells themselves on the one hand, and between tumor and host cells on the other is not totally understood and remains one of the main next frontiers in oncology. Such an approach will benefit from an ecological perspective, a viewpoint that will be considered later in this narrative.

Evolvability is an ecological term that reflects the adaptive ability of a species to generate and maintain a heritable phenotypic diversification to prevent extinction [4]. At least theoretically, metastasis may be conceived as an example of tumor evolvability achieved by genetic and epigenetic modifications that confer migratory abilities to cells to escape to more “favorable” ecosystems. A significant amount of studies are being published focusing on the intricate genomic/epigenomic complexity of this issue, unveiling the multiple routes that enable malignant cells to acquire locomotion capacities. However, very few studies have analyzed the collective pressures and cell-to-cell interactions that may explain the reasons for which malignant cells decide to migrate far away. This new approach implies considering tumors as a sort of social dysfunction [5] and analyzing malignant cells and their microenvironment from an ecological viewpoint [6].

In this perspective, we approach the metastatic process from genomic/epigenomic, ecological, and sociological perspectives. On the one hand, we review some of the last salient genomic/epigenomic findings related to the acquisition of metastatic capacities focusing on a paradigmatic example of intratumor heterogeneity and metastatic selection: The clear cell renal cell carcinoma (CCRCC). Tumor/non-tumor cell interactions (endothelial cells, tumor-infiltrating lymphocytes, and cancer-associated fibroblasts) are also mentioned as a substantial part of the adaptive processes leading to tumor migration. Finally, we analyze cell-to-cell interactions using a game theory approach and hypothesize that metastasis may be simply a specific response of a subset of tumor cells. Such response would consist of searching for collective stability far away from the primary tumor to improve their collective wellness and prevent extinction.

## 2. Molecular Approach

The molecular characterization of CCRCC has been largely analyzed [7]. Clonal and sub-clonal evolutions generate in some of these carcinomas a high variability of metastatic patterns. While some tumors typically develop only one metastatic clone, others are able to develop several ones with specific capacities and topographic affinities [8]. CCRCC is a paradigmatic example of intratumor heterogeneity (ITH) [9]. Actually, some deterministic constraints with prognostic impact [10] and two different metastatic patterns [11] have been identified in this tumor. A rule-based classification system supported by unsupervised clustering comparing different genomic, histological, and clinical data has shown up to seven evolutionary subtypes [10]. Multiple clonal drivers, *BAP-1* driven, and *VHL* wild type displayed a punctuated evolution and accelerated clinical progression. On the other hand, CCRCC related to *PBRM-1* gene dysfunction (PBRM-1 → SETD2, PBRM-1 → PI3K, PBRM-1 → SCNA) pursued a less aggressive clinical progression and a branched evolution. The seventh subtype detected, the *VHL* mono-driver, showed a monoclonal structure without additional driver mutations.

Losses of 9p and 14q are molecular alterations linked with metastatic ability in CCRCC [11]. Tumors that follow a punctuated model of evolution display high chromosomal complexity but low ITH. Here, a single clone with high fitness fixes early in its evolution and occupies a significant percentage of the tumor mass following a Darwinian pattern. These cases develop multiple metastases early in their evolution (rapid progression). The branching pattern, however, is also a Darwinian example of tumor evolution, but in this case, there is high clonal and sub-clonal diversification. As a result, chromosomal complexity is low, but ITH is high. Branching-type tumors develop few metastases late in their evolution (attenuated progression).

Hypoxia promotes metastases, but this factor is not homogeneously distributed across the tumor. Actually, a tumor spatial specialization pattern has been very recently detected in CCRCC. Tumor areas with metastatic abilities are located in the tumor interior in a multi-regional analysis of 79 cases [12]. Tumor necrosis and higher Ki-67 index, histological grade, and chromosomal complexity were detected in the tumor center, where microenvironmental pressures and the struggle for survival are supposedly higher due to higher levels of hypoxia.

The metastatic capacity is associated with the acquisition of a stem cell-like phenotype because of the evolvability of a subset of tumor cells, the so-called metastasis-initiating cells [13]. Metastatic dissemination (local invasion, intravasation, blood and/or lymphatic circulation, extravasation and extravascular spread), dormancy, immune evasion, organ colonization, and local tropism development conform to the stepwise process followed by this special subset of traveling tumor cells. This concatenated sequence of events needs a dynamic epigenetic remodeling to adapt these cells to the ever-changing environments. Abnormal DNA methylation (methyl-binding proteins, post-translational histone modifications, miRNAs, lncRNAs, RNA methylation) is one of the epigenetic mechanisms most extensively studied and an actionable target for future treatments [14].

## 3. Microenvironmental Context

A question arises at this point: How many times can tumor cells migrate? In other words, can a metastasis metastasize? The answer is yes, as it has been demonstrated in genomic studies analyzing the clonality patterns of the widespread tumor seeding in renal, prostate, breast, pancreatic, and colo-rectal cancers [15,16]. Such complex polyclonal dissemination seems a cornerstone in the development of prostate cancer clones resistant to castration, as demonstrated by Gundem et al. [15].

The relationship between tumor cells and their microenvironment is crucial in cancer progression and metastatic process. Immune cells, neovascular endothelia, and fibroblasts are the best-studied tumor-associated cells. These stromal elements dynamically interact between and with tumor cells. In addition, this interplay is becoming a promising therapeutic target, with immune checkpoint blockade (PD-1, PD-L1, CTLA-4) and anti-angiogenesis as two of the most representative examples of precision therapies in CCRCC [17] and other neoplasms. In fact, the combination of both treatments in a subset of these patients has a positive effect based on the synergic actions of antiangiogenic and checkpoint blockers [17].

PD-1 and its ligand PD-L1 (B7-H1) blockade has been implemented in clinical practice as a promising therapeutic strategy in CCRCC and other tumors, but the immunohistochemical selection of candidates, with different antibodies and cut-offs, is still controversial [18]. By contrast, the soluble fraction of PD-L1 has been associated with prognosis and metastatic status in these tumors [19]. Immune checkpoint blockade seems especially useful in the so-called inflamed tumors, a group of aggressive CCRCC with abundant tumor-infiltrating lymphocytes linked to *BAP-1* gene inactivation and tendency to early metastases [20]. On the other hand, B7-H3, another immune regulator of the B7 family, has been directly implicated in the metastatic process [21], expanding the list of candidates for checkpoint blockade in the clinical practice.

CCRCC is a neoplasm associated with *VHL* gene malfunction that creates a pseudo-hypoxic status in neoplastic cells. This condition induces the initialization of the VEGF cascade [9], ending in neoangiogenesis. A new definition of microvessel density in CCRCC has been recently published [22]. These authors consider the sum of classical microvessel density and vasculogenic mimicry as the total microvessel density and correlate this sum with prognosis [22]. Since these neoplasms are highly vascularized, the antiangiogenic tyrosine-kinase inhibitors have had a relevant role in the therapeutic armamentarium, particularly in metastatic patients [23].

Not all CCRCC, however, display the same degree of angiogenesis. Predominantly angiogenic examples are typically associated with *PBRM-1* gene inactivation [20] and show a pancreatic tropism [24]. Interestingly, pancreatic metastases in angiogenic-type low-grade CCRCC have been detected significantly later in the tumor evolution compared with metastases in the rest of the sites [11].

Cancer-associated fibroblasts (CAF) are essential elements in the tumor microenvironment [25]. The interplay between tumor cells and CAF activates the epithelial-to-mesenchymal transition process providing a stem cell phenotype to epithelial tumor cells, thus enabling tumor cell migration [26]. In this interaction, CAF produce fibroblast activation protein-α (FAP), a protein associated with biological aggressiveness [27] and early metastases [28] in CCRCC. Other proteins produced by CAF, however, also contribute to tumor progression [29]. Coercive feedback signaling from cancer cells to CAF does exist, thus assuring the maintenance of FAP production and perpetuating the loop [26].

## 4. Sociological Approach

Game theory [30] is a branch of applied mathematics initially applied to economics that investigates interactions between individuals. It has been used to examine complex problems in biology [31] and, more recently, in oncology [32]. Considering cancer as a social dysfunction [5] in which cells interact following different strategies, game theory has opened up new perspectives to understand cancer dynamics.

The Hawk–Dove game (Figure 1) is a classical model in biology [31]. It assumes a population whose members have bilateral encounters to divide a resource (v), with some cost (c) associated with fight (c > v). In every encounter, each individual can behave as a “hawk” and escalate to a fight or as a “dove” and back down. Therefore, if one individual acts as a hawk while the other acts as a dove, the “hawk” gets the resource (v) and the dove gets nothing (0). If two “hawks” meet, there is a fight, the winner receives the resource (v) and the other faces the cost of the fight (−c). On average, the two hawks receive (v − c)/2. If two individuals act as dove, they share the resource. On average, the two doves get v/2.

The concept of evolutionarily stable strategy (ESS) is used to solve the game. ESS in game theory captures the resilience of a strategy against another in the sense that it assumes that most members of the population play the ESS but a small proportion of members, here termed mutants, chooses a different strategy. In this context, each mutant’s expected payoff is smaller than the expected payoff of an individual who plays the ESS. Consequently, mutants are driven out of the population. As a consequence, the Hawk–Dove game has a unique ESS, which consists of a fraction, v/c, of the population which plays hawk while the remaining population plays dove.

This description is appropriate for analyzing the interactions between cancer cells and cancer-associated fibroblasts in CCRCC [25] that we have previously mentioned in this narrative. The resource, in this case, is the diffusible factor fibroblast activation protein-α (FAP) that improves cancer fitness, while the cost is the effort of producing and excreting it to the medium.

It is thus theoretically possible to predict that a homogeneous neoplasm will achieve its ESS. However, there is no such thing as a perfectly homogeneous tumor in real life, so a heterogeneous population must be considered. In the heterogeneous Hawk–Dove game [33], the population is divided into two types. In consequence, each individual conditions his/her action depending on his/her opponent’s type. In the heterogeneous game, the previously described ESS is not stable any longer. Therefore, the game will have two ESSs. Each ESS is characterized by one of the two types of individuals being discriminated. In such a context, one type of individual receives a higher payoff than the other. Furthermore, the payoff of the discriminated individuals in the heterogeneous game is smaller than the payoff obtained in the homogeneous one.

However, there is an unanswered question in oncology: Why does a malignant tumor metastasize? From an ecological viewpoint, the goal of cell migration is tumor survival. From a sociological perspective, however, it can be hypothesized that the ultimate common endpoint of neoplastic cells that migrate far away is to achieve a higher payoff elsewhere. Here, the set of discriminated cells in the heterogeneous population that governs the primary tumor might develop metastatic abilities just to escape to a friendlier environment where a higher payoff is initially possible, at least theoretically.

## 5. Conclusions

Tumor spatial specialization with metastasizing subclones located in the tumor interior, the demonstrated ability of metastases to metastasize, and the sociological tumor cell interactions unveiled by the Hawk–Dove game reinforce the storyline of this perspective in the sense that the search for a better environment by tumor cells is a constant event in malignant tumors.

## Figures and Tables

**Figure 1 cancers-13-03653-f001:**
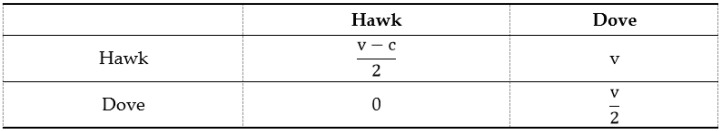
Hawk–Dove game matrix. Each of the two individuals chooses one strategy. The matrix summarizes the payoffs of the four possible results for the individual choosing the row strategy playing against an individual choosing the column strategy (v = resource, c = cost).

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
