# Peer review of "Metastasis, an Example of Evolvability"

_cancers, 2021, doi:10.3390/cancers13153653_

Round 1

Reviewer 1 Report

The paper entitled "Metastasis, an example of evolvability" focused the attention on the game theory applied to cell encounters within a tumor, providing a sociological perspective of the possible behaviors of individuals (cells) in a collectivity. This combined approach provides a more comprehensive understanding of the complex rules governing a neoplasm.

The manuscript is really well written and of interest for the audience.

Minor comments:

  • The Authors should provide the expand forms for all acronyms, including gene and protein acronyms, thorugh the text when they first appear.
  • Could the Authors mention and discuss the possibility of the application of the game theory also for metastasis developed within other tumors?

Author Response

Thank you for your comments.

Most of the acronyms are from genes or proteins and are called by the acronym throughout the literature. We have pormenorized VEGF (vascular endothelial growth factor).

Game theory is a tool that can applied to any type of tumor, since all tumors are without exception, communities of individuals with different interests and pressures. We have mentioned one specific type of cancer (clear cell renal cell carcinoma, CCRCC) because we have more experience on it and because is a paradigmatic example of a heterogeneous neoplasm, but any other one could also serve as an example.

Reviewer 2 Report

This article lacks a clear direction on how to investigate of cancer metastasis.  In addition, I don’t agree the authors mentioning that the intricate interactions between cancer and host cells is largely unknown (introduction) and metastasis only ends when the tumor kills the patients (conclusion). The most cancers are curable or can be treated like chronic diseases.

Author Response

Thank you for your comments. Yes, the manuscript is just a "Perspective", a call for oncologists, pathologists, and other scientists, mathematicians for example, to work together. At the end, cancer is much more than a genetic disease.

Yes, the sentence in the introduction "is largely unknown" has been changed by "is not totally understood", what is softer and for sure more realistic.

Yes, the last sentence in the conclusion is ellucubrative and has been removed.

I agree that many cancers are curable or at least treated as chronic diseases. However, a significant number of them are aggressive and cancer-related mortality is still high.

Reviewer 3 Report

The authors have reviewed and explained the metastatic process in two concepts, molecular and sociological approaches. Although many scientists and clinicians would be aware of this concept and this concept is not original, the authors have reviewed the related studies and presented the concepts in well organized fashion. I believe that this study will again remind many readers refreshing their concept of metastasis and provide wider perspectives. 

Author Response

Thank you for your comments. Yes, we have connected two different approaches that usually go separately. The manuscript is intended as a "perspective", that is, a call for integrating different viewpoints to sum evidence in the topic. Pathologists and oncologists must be familial with approaches coming from other sciences. At the end, cancer is more than a genetic disease.

Round 2

Reviewer 2 Report

I maintained my original decision.

Author Response

Thank you for your suggestion. We agree. Since the manuscript is a Perspective and has not a classical discussion, we have included the following two sentences in the introduction (they appear in red in the revised version) to stress, from the very beginning, the importance of the reviewer 2's comment:

Modern treatment modalities are obtaining longer survivals and even cure in a significant percentage of patients with cancer transforming the disease in a chronic-like condition. However, cancer still remains today as the leading cause of death in Western countries. 

I hope it works now.